# Human Mutated *MYOT* and *CRYAB* Genes Cause a Myopathic Phenotype in Zebrafish

**DOI:** 10.3390/ijms241411483

**Published:** 2023-07-14

**Authors:** Elena Cannone, Valeria Guglielmi, Giulia Marchetto, Chiara Tobia, Barbara Gnutti, Barbara Cisterna, Paola Tonin, Alessandro Barbon, Gaetano Vattemi, Marco Schiavone

**Affiliations:** 1Department of Molecular and Translational Medicine, Zebrafish Facility, University of Brescia, 25123 Brescia, Italy; 2Department of Neurosciences, Biomedicine and Movement Sciences, Section of Clinical Neurology, University of Verona, 37134 Verona, Italy; 3Department of Neurosciences, Biomedicine and Movement Sciences, Section of Anatomy and Histology, University of Verona, 37134 Verona, Italy

**Keywords:** myotilin, αB-crystallin, myofibrillar myopathy, zebrafish model

## Abstract

Myofibrillar myopathies (MFMs) are a group of hereditary neuromuscular disorders sharing common histological features, such as myofibrillar derangement, Z-disk disintegration, and the accumulation of degradation products into protein aggregates. They are caused by mutations in several genes that encode either structural proteins or molecular chaperones. Nevertheless, the mechanisms by which mutated genes result in protein aggregation are still unknown. To unveil the role of myotilin and αB-crystallin in the pathogenesis of MFM, we injected zebrafish fertilized eggs at the one-cell stage with expression plasmids harboring cDNA sequences of human wildtype or mutated *MYOT* (p.Ser95Ile) and human wildtype or mutated *CRYAB* (p.Gly154Ser). We evaluated the effects on fish survival, motor behavior, muscle structure and development. We found that transgenic zebrafish showed morphological defects that were more severe in those overexpressing mutant genes. which developed a myopathic phenotype consistent with that of human myofibrillar myopathy, including the formation of protein aggregates. Results indicate that pathogenic mutations in myotilin and αB-crystallin genes associated with MFM cause a structural and functional impairment of the skeletal muscle in zebrafish, thereby making this non-mammalian organism a powerful model to dissect disease pathogenesis and find possible druggable targets.

## 1. Introduction

The term myofibrillar myopathy (MFM) refers to a group of clinically and genetically heterogeneous neuromuscular disorders sharing common histological features, such as Z-disk dissolution, myofibrillar degeneration, and the accumulation of myofibrillar degradation products [1,2,3,4]. MFM represents the paradigm of protein aggregates myopathies (PAM) characterized by protein misfolding and aggregation in skeletal muscle tissue that clinically result in muscle weakness, frequently associated with non-skeletal muscle symptoms [5]. The MFM clinical spectrum is wide and mainly consists of progressive proximal and/or distal limb muscle weakness; limb–girdle and scapuloperoneal phenotypes can be observed, as well as the involvement of hand, facial, and pharyngeal muscles [1,2,3,4]. Cardiomyopathy, peripheral neuropathy, respiratory impairment/failure, and cataracts are frequently associated conditions [1,2,3,4]. The diagnosis is established via muscle biopsy, which shows abnormal fibers containing amorphous material of irregular shape and size that are positive for several proteins, including αB-crystallin, desmin, and myotilin, as the main morphological hallmark. [1,2,3,4]. MFMs are usually transmitted as an autosomal dominant trait; however, X-linked, autosomal recessive and sporadic (caused by *de novo* mutations) cases have been described [1,2,3,4]. Causative mutations have been identified in several genes encoding proteins of the Z-disk structure or protein chaperones [1,2,3,4,5,6]. To date, most genetically diagnosed patients harbor pathogenic variants in one of the following genes: desmin (*DES*), αB-crystallin (*CRYAB*), myotilin (*MYOT*), Z-band alternatively spliced PDZ-containing protein (*LBD3/ZASP*), Bcl2-associated athanogene-3 (*BAG3*), and filamin C (*FLNC*) [1,2,3,4,5,6]. Despite the identification of several disease-linked genes, the pathogenic mechanisms leading to protein aggregation and, ultimately, to muscle fiber impairment have not been elucidated yet [7]. Thus, the use of animal models is critical for investigating the molecular basis of the disease and developing new therapeutic strategies to treat these progressive and disabling muscle disorders. Several animal models of MFM, based on *DES*, *ZASP*, *BAG3* and *FLNC* human gene mutations, have been described [8]. Conversely, few models expressing rare human mutations in *CRYAB* and *MYOT* exist [9,10,11,12,13,14,15,16].

Thanks to their transparency, easy handling, easy genetic manipulation, and highly conserved genetic and molecular mechanisms that control muscle development and function, zebrafish (*Danio rerio*) is a vertebrate model widely used to study human myopathies, such as muscular dystrophies, congenital myopathies, myofibrillar myopathies, and cardiomyopathies [15,17,18,19,20,21,22]. Aminoacidic sequence identity of MFM proteins is high, especially in functional domains, ranging from 41% (BCL2-associated athanogene 3, myotilin and αB-crystallin) to 97% (desmin, filamin C, Z-band alternatively spliced PDZ-containing protein). Several studies based on the inactivation of MFM gene function using morpholino-modified antisense oligonucleotide-mediated gene knock-down demonstrated that the targeted inactivation of MFM genes in zebrafish led to a severe myopathy characterized by sarcomere degeneration without pathological protein aggregation [15]. MFM gene mutations, however, do not usually result in a complete loss of function but rather are likely to cause a dominant negative effect.

Here, we report the first attempt to characterize in zebrafish the effects of the rare human p.Gly154Ser and p.Ser95Ile missense mutations in the αB-crystallin and myotilin proteins, respectively. Both pathogenic variants have been reported in patients whose biopsies showed myopathic features resembling those of myofibrillar myopathy [4,23,24,25].

The *CRYAB* G154S mutation was chosen because patients with this missense mutation do not show cataracts and/or cardiomyopathy. This mutation is, therefore, suitable to evaluate the MFM phenotype at the level of the skeletal muscle. Moreover, zebrafish lines (or other animal models) that express the αB-crystallin mutants Gly154Ser have never been generated.

*MYOT* S95I mutation was selected since it is the only single-point missense variant located in the exon 2, which lies in the domain necessary for the interaction of the protein with alpha-actinin, an actin crosslinking protein, making the mutant zebrafish attractive to study the role of myotilin in the pathogenesis of the disease through the impairment of the correct assembly of the contractile apparatus. In addition, zebrafish lines or other animal models carrying the p.Ser95Ile variant have never been generated.

However, both residues (p.Gly154 for αB-crystallin and p.Ser95 for myotilin) are not conserved in zebrafish protein sequences. Thus, we conducted a pilot study to explore the possibility of replicating an MFM-like phenotype in zebrafish by inducing the transient expression of human mutated genes [26]. Our results demonstrate zebrafish as a valid model to study human MFM due to specific mutations in *CRYAB and MYOT* genes and suggest the future generation of stable mutant lines useful to go towards precision medicine [27].

## 2. Results

### 2.1. Patients with Myofibrillar Myopathy

We identified 54 patients with MFM who were clinically evaluated and underwent open muscle biopsy at the Neuromuscular Center of our Institution between January 1995 and December 2021. The cohort included 48 unrelated families and three pairs of brothers. The diagnosis of MFM was established according to the diagnostic criteria provided by Schröder and Schoser [2]. Pathogenic mutations in MFM-related genes, including desmin (*DES*), αB-crystallin (*CRYAB*), myotilin (*MYOT*) and filamin C (*FLNC*), were identified in 10 of the 54 MFM cases (19%) [4]. The most frequent mutation in our cohort was in *MYOT*, which was found in five patients with genetically determined MFM, followed by *DES* (three out of ten); only one patient had a pathogenic variation in *CRYAB* and the last one was in *FLNC*. None of our 54 patients with MFM harbored pathogenic mutations in the LIM domain-binding 3 (*LDB3/ZASP*) and BCL2-associated athanogene 3 (*BAG3*) genes. Representative histological (Figure 1), immunohistochemical (Figure 2), and ultrastructural (Figure 3) findings from the muscle of patients with genetically determined MFM and control subjects are shown in Figure 1, Figure 2 and Figure 3.

### 2.2. In Vivo Results in Zebrafish

#### 2.2.1. Dose–Response Assay

To study both *MYOT* and *CRYAB* function and the effects of rare human mutations *MYOT* S95I and *CRYAB* G154S, we injected wildtype zebrafish embryos at the one-cell stage with eukaryotic expression plasmids harboring cDNA of either human wildtype or mutated genes. First, we checked for the optimal concentration of plasmids that show the lowest toxicity in zebrafish embryos. After four independent injections, we chose a concentration of 25 μM. After the injection of the four plasmids, we assessed the effects of wildtype and mutated *hMYOT* and *hCRYAB* genes on motor behavior, muscle structure, BMP signaling pathway and fish survival.

#### 2.2.2. Analysis of Motor Behavior and Muscle Structure Development

To check the effects on motor behavior, we analyzed both the number of spontaneous coiling events (tail flips) performed in 30 s using zebrafish embryos at 24 hpf and the ability of zebrafish larvae at 72 hpf to move after a touching stimulus (touch-evoked escape response). We found a significant reduction in tail flips in both *hMYOT* S95I and *hCRYAB* G154S-injected zebrafish embryos at 24 hpf compared to both wildtype siblings either not injected or injected with wildtype genes (Figure 4A). A significant reduction in the touch-evoked escape response score was also observed in *hMYOT* S95I and *hCRYAB* G154S-injected zebrafish embryos at 48 hpf, even if embryos overexpressing wildtype genes showed motor impairment, as previously reported [28,29] (Figure 4B).

These results fit well with those of birefringence assay, which analyzes the integrity of muscle structure by taking advantage of muscle anisotropy, i.e., the ability of normal muscle fiber to refract the light polarized by polarizing filters as described in the materials and methods section [30]. We analyzed the birefringence of wildtype embryos either not injected or injected with the different plasmids at 48 hpf (a developmental stage in which the zebrafish muscle fibers are fully organized) [31,32]. We observed abnormal birefringence in 34% of embryos injected with *hMYOT* S95I and in 43% of embryos injected with *hCRYAB* G154S, suggesting the presence of severe muscle damage induced by these human mutations (Figure 5A). Fewer embryos injected with *hMYOT* WT and *hCRYAB* WT also showed an altered birefringence phenotype (17% and 26%, respectively), highlighting the importance of both genes in muscle structure maintenance (Figure 5A).

To check if defects in muscle structure are due to muscle differentiation problems, we injected 25 μM of our four plasmids in fertilized eggs at the one-cell stage obtained from incrosses of stable transgenic line expressing GFP under the control of sequence responsive elements recognized by Id1, which is a specific effector of canonic BMP signaling pathway [33]. Indeed, the BMP signaling pathway has been shown to play a key role during the first stages of muscle development and differentiation in zebrafish [34]. Some studies demonstrated that BMP signaling increases during the early stages of muscle development, inducing the proliferation of muscle progenitor cells and initial muscle hypertrophy by inhibiting both TGFβ/activin and Shh pathways [31,35]; other studies demonstrated that high BMP signal triggers muscle atrophy [34,36,37]. Even if opposite conclusions on BMP and muscle maintenance have been drawn in different studies, it is clear that this pathway takes part in balancing hypertrophy and atrophy to maintain the integrity of muscle structure is possible [38,39,40]. In addition, the activation of BMP signaling has been shown to cause myopathy with reduced muscle mass, central nuclei in several muscle fibers and increased fat deposition in Fbn2 null mice during early postnatal muscle development, suggesting that the BMP signaling pathway may have different effects depending on microenvironment [41]. Moreover, *CRYAB* is regulated at the level of transcription by the BMP signaling pathway, suggesting crosstalk between BMP and αB-crystallin [42].

We observed different effects of human mutated *CRYAB* and *MYOT* genes on BMP pathway activity. An impairment of BMP was documented by a dramatic decrease in GFP fluorescence in embryos injected with *hMYOT* S95I (Figure 5B); conversely, an increase in BMP activity was observed in those injected with *hCRYAB* G154S. These findings are in agreement with the muscle atrophy observed in *hMYOT* S95I injected embryos and the muscle hypertrophy reported in *hCRYAB* G154S. We found a similar alteration in terms of BMP activity in embryos injected with wildtype genes, suggesting that the observed changes in BMP activity are due to gene overexpression (more evident in *CRYAB* than in *MYOT*).

#### 2.2.3. Analysis of Fish Survival

We also determined the effects of human genes on fish survival by performing a short Kaplan–Meier analysis recording the number of dead embryos every 24 h until 7 dpf. We observed a progressive decrease in fish survival (Figure 6A,B) in all injected conditions, pointing to a possible negative effect of gene overexpression. This effect is more visible with *CRYAB* than with the *MYOT* gene.

#### 2.2.4. Transmission Electron Microscopy and Immunofluorescence Analysis

To further characterize the myopathic phenotype of zebrafish embryos injected with human mutations, muscle ultrastructure and protein aggregation were analyzed using (i) transmission electron microscopy and (ii) whole-mount immunofluorescence by incubating pools of at least seven embryos for each condition with antibodies specific for αB-crystallin and myotilin. Ultrastructural analysis revealed significant disorganization of sarcomere structures in both *hMYOT* S95I and *hCRYAB* G154S injected zebrafish embryos at 48 hpf (Figure 7I–L) compared to both wildtype siblings either not injected or injected with wildtype genes (Figure 7A–H), recapitulating a feature of human MFM muscle. Interestingly, we observed gene-specific ultrastructural abnormalities. Embryos injected with *hMYOT* S95I showed an irregular Z-disk morphology that appears less electron-dense and zigzagged (Figure 7I,J). Large areas devoid of myofibrils containing only a few organized sarcomeres and stressed fiber-like structures and less compact myofibrils were found in embryos injected with mutated *hCRYAB* (Figure 7K,L). In addition, whole-mount immunofluorescence analysis, performed with antibodies specifically recognizing human protein isoforms, and the relative embryos imaging at 48 hpf with both Axiozoom fluorescent microscopy (Figure 8) and lightsheet Z1 microscopy (Figure 9) clearly showed the correct expression of human proteins at the level of skeletal muscle fibers in embryos injected with wildtype *hMYOT* and *hCRYAB* isoforms. Both imaging analyses documented the presence of a few differently shaped dots with high fluorescence intensity in muscle fibers of embryos injected with both wildtype and mutant *hCRYAB* and *hMYOT*. Fluorescent dots in embryos injected with wildtype *hCRYAB* and *hMYOT* are not linked to an MFM-like phenotype, unlike those observed in embryos injected with mutant *hCRYAB* and *hMYOT* that show well-defined, highly fluorescent spots similar to those observed in human muscle patients. This result strongly supports the one obtained by TEM analysis (Figure 7), suggesting the possible formation of myofibrillar aggregates, which are typical features of MFM in embryos injected with *hMYOT* S95I and *hCRYAB* G145S.

## 3. Discussion

In this pilot study, we explored the possibility of generating the first zebrafish models of MFM expressing two different pathogenic variants, *CRYAB* G154S and *MYOT* S95I, found in MFM patients [4,23,24,25].

The myotilin gene (*MYOT*) maps to the 5q31.2 genome region and encodes a 57 kDa sarcomeric Z-disk protein consisting of 498 amino acids, which is expressed in skeletal and cardiac muscle and peripheral nerves [43]. Myotilin is characterized by the presence of an N-terminus that contains a serine-rich region responsible for its interaction with α-actinin and of two immunoglobulin-like domains at the C-terminus that are necessary for protein homodimerization and the binding of actin and filamin C [43,44]. The protein plays a crucial role in the correct assembly of the contractile apparatus through cross-linking actin filaments into tightly packed bundles at Z-disk and protects thin filaments from depolymerization [43]. Mutations in *MYOT* have been originally associated with different disorders separated according to the clinical and histological findings into three main categories including autosomal dominant limb–girdle muscular dystrophy type 1A (LGMD1A), myofibrillar myopathy (MFM), and spheroid body myopathy [45,46]. Subsequently, LGMD1A has been classified as a form of MFM [47]. At present, diseases caused by mutations in *MYOT* are considered as a single entity, namely myotilinopathies (OMIM #609200), characterized by a continuum of phenotypic and pathological manifestations [47]. These disorders are usually transmitted as an autosomal dominant trait and missense mutations in *MYOT* account for approximately 10% of genetic variations in patients with MFM [45,48]. *MYOT* variants are mainly located in exon 2 encoding the serine-rich domain and their pathogenic mechanism remains largely unknown [47]. The dominant myofibrillar myopathy causing *MYOT* S95I mutation was reported in one affected individual with myotilinopathy and is the only single-point missense variant located in the exon 2 that lies in the domain necessary for the interaction of the protein with alpha-actinin [25]. This patient showed a slowly late-onset disease course without cardiac involvement and histopathological features consistent with the diagnosis of MFM [25].

The human *HSPB5* gene (*CRYAB*), located on chromosome 11q23, encodes αB-crystallin a 175 amino acid protein with a molecular mass of 20 kDa [49]. The protein, which is highly expressed in tissues with elevated rates of oxidative metabolism, including cardiac and skeletal muscle, consists of a conserved central domain called the “α-crystallin domain” (60–150 aa), the flanking N-terminal region, and the C-terminal region [50,51,52]. In adult skeletal muscle, αB-crystallin, a small heat shock protein (HSP) that acts as a molecular chaperone binding misfolded and denatured proteins to prevent their aggregation, is located at the level of the Z-disk and is associated with actin and titin [52,53]. Given the important functions of protein in muscle differentiation, protein degradation and cytoskeletal stabilization, it is not surprising that mutations in *CRYAB* result in skeletal and cardiac disorders such as myofibrillar myopathy and various forms of cardiomyopathy including dilated and restrictive [52,54]. To date, different myofibrillar myopathy causing *CRYAB* mutations have been identified, almost all demonstrating autosomal dominant inheritance and located in the highly conserved α-crystallin domain or in the flexible C-terminal segment [52,54]. Heterozygous missense mutation p.(Gly154Ser) in the *CRYAB* gene, a variation that maps on the C-terminal region of the protein, has been associated with a late-onset progressive distal myopathy without cardiac involvement or isolated cardiomyopathy [23,24]. The same p.(Gly154Ser) variant was detected in the only case of our series carrying a causative mutation in *CRYAB* [4]. This patient showed an early adult-onset muscle weakness involving mainly distal upper limbs without cardiac impairment [4].

The molecular mechanisms dysregulated in skeletal muscle from patients with mutations in genes encoding for αB-crystallin and myotilin, which are representative of the structural and chaperone proteins involved in MFM, are still under investigation, although their common pathological features are well characterized. MFMs are still incurable diseases, and the attempt to move towards precision medicine to find the appropriate pharmacological treatment for a single patient with a specific mutation makes it necessary to develop simple animal models as tools to both study pathogenic mechanisms and perform fast, high-throughput drug screening.

Up to now, different model organisms, including transgenic mice carrying the *MYOT* p.T57I or the *CRYAB* p.R120G variants and loss-of-function αB-crystallin or myotilin zebrafish mutant lines, have been generated in the attempt to uncover the molecular events that lead from gene mutation to disease [9,10,11,12,13,14,15,16]. However, the associated phenotypes among the animal models have focused on the heart and/or lens [10,11,12,13,16] or do not sufficiently reflect the disease phenotype observed in the muscle of patients [9,14]. Moreover, a gain of function or dominant interaction with wildtype protein rather than a complete loss of gene function is thought to be responsible for these myopathies.

In this pilot study, we chose zebrafish because of its numerous advantages as embryo transparency, quick development, and a high degree of both nucleotide and aminoacidic sequence identity in genes and proteins involved in muscle development and diseases. Regarding the proteins involved in MFM, the range of sequence identity is from 41% (myotilin, αB-crystallin, BCL2-associated athanogene 3) to 97% (desmin, filamin C, Z-band alternatively spliced PDZ-containing protein).

Unfortunately, despite a high degree of aminoacidic sequence identity between human and zebrafish proteins, both residues (G154 for αB-crystallin and S95 for myotilin) are not conserved. Thus, to understand whether these human mutations can induce a skeletal muscle phenotype in zebrafish, we transiently overexpressed both wildtype and mutated human genes by injecting plasmids harboring *hCRYAB* WT, *hCRYAB* G154S, *hMYOT* WT and *hMYOT* S95I cDNA sequences in wildtype zebrafish embryos at the one-cell stage.

We demonstrated that the overexpression of *CRYAB* G154S and *MYOT* S95I in muscle tissue results in a clearly myopathic phenotype characterized by a significant reduction in myofiber density, a marked disorganization of sarcomeres and the formation of granular protein aggregates, all of which are characteristic histopathological features of human MFM [2]. Interestingly, mutations in each of the two genes caused distinct structural abnormalities, strengthening the evidence for different mechanisms by which the two groups of proteins (structural and chaperone) cause MFM disease. Moreover, we found that the overexpression of *CRYAB* and *MYOT* wildtype could impair zebrafish development and lead to the formation of aggregates, which appear not to be correlated with a myopathic phenotype as demonstrated by electron microscopy analysis on muscle tissue. This result confirms that both proteins contribute to the development of structural and functional muscle tissue, as previously demonstrated [28,29].

This was also confirmed by altered BMP signaling activity, which role in muscle development is still debated since several studies demonstrated the high variability of this signaling pathway in different developmental stages of muscle tissue [31,34,35,36,37]. We found an increase in BMP activity in embryos injected with *hCRYAB* G154S, suggesting muscle hypertrophy, which could be a compensatory response to the αB-crystallin defects as it is a target of BMP signaling [42]. As expected, we observed a decrease in BMP activity in embryos injected with *hMYOT* S95I, showing muscle atrophy due to the disruption of muscle sarcomere organization linked to myotilin dysfunction. As both proteins play a major role both in keeping the correct organization of muscle structure and in muscle tissue development, the overexpression of wildtype isoforms leads to unbalanced BMP activity confirming its possible untoward effect.

Finally, all our results demonstrated the importance of a balanced concentration of these proteins to maintain the integrity of muscle structure. This is the reason why zebrafish injected with human wildtype proteins showed altered phenotypes. However, the injection of human mutated genes led to a myopathic phenotype consistent with a myofibrillar myopathy, such as motor impairment, defects in muscle fiber assembly and development, higher mortality compared to not injected embryos, and the presence of protein aggregates in damaged muscle fibers.

This pilot study validates zebrafish as a powerful tool to study human MFM due to specific mutations in *CRYAB* and *MYOT* genes, setting the stage for the generation of stable mutant lines useful to go towards precision medicine.

## 4. Materials and Methods

### 4.1. Histology, Histochemistry, Electron Microscopy and Immunocytochemistry

All muscle biopsies were performed for diagnostic purposes after written informed consent. The study was conducted in accordance with the Declaration of Helsinki and approved by the local ethical board of the Department of Neurological, Neuropsychological, Morphological, and Movement Sciences, University of Verona, Verona, Italy.

Muscle samples were snap-frozen in liquid-nitrogen-cooled isopentane. Serial 8-μm-thick cryosections were stained with standard histological and histochemical methods, including haematoxylin and eosin (H&E), modified Gomori trichrome, adenosine triphosphatase (ATPase, pre-incubation at pH 4.3, 4.6 and 10.4), succinate dehydrogenase (SDH), cytochrome c oxidase (COX), reduced nicotinamide adenine dinucleotide (NADH), periodic acid-Schiff (PAS) with diastase digestion, Sudan black and acid phosphatase [4].

A small fragment of muscle tissue was fixed in 2.5% glutaraldehyde in 0.1 M phosphate buffer, pH 7.4, at 4 °C overnight, post-fixed with 1% osmium tetroxide and 1.5% potassium ferrocyanide at RT for 2 h, dehydrated with acetone and embedded in Spurr resin. Semithin sections were stained with toluidine blue and PAS. Ultrathin sections were stained with uranyl acetate and observed with a Philips Morgagni transmission electron microscope (FEI Company Italia Srl, Milan, Italy) operating at 80 kV and equipped with a Megaview II camera for digital image acquisition [55].

Double immunofluorescence was carried out on 9-μm-thick transverse serial muscle sections with antibodies to desmin, αB-crystallin and myotilin in combination with biotinylated Maackia amurensis lectin-II (MAL-II, Vector Laboratories, Burlingham, CA, USA) that labels the sarcolemma of the muscle fiber. Immunofluorescence was performed as previously described [56,57]. Controls were muscle biopsies from subjects who were ultimately deemed to be free from muscle diseases.

### 4.2. Genetic Investigations

Genomic DNA was extracted from venous blood using standard methods. All MFM patients underwent genetic testing by Sanger sequencing for mutations in the following genes: Desmin (*DES*, NCBI Reference Sequence: NG_008043.1), Myotilin (*MYOT*, NCBI Reference Sequence: NG_008894.1), Crystallin, alpha B (*CRYAB*, NCBI Reference Sequence: NG_009824.1), LIM domain-binding 3\(*LDB3*, NCBI Reference Sequence: NG_008876.1), BCL2-associated athanogene 3 (*BAG3*, NCBI Reference Sequence: NG_016125.1) and Filamin C (*FLNC*, NCBI Reference Sequence: NG_011807.1).

### 4.3. RNA Isolation from Human Skeletal Muscle Tissue

Total RNA was isolated from the skeletal muscle of a subject that, after all the analysis, was deemed to be free of skeletal muscle diseases. RNA was extracted using the TRIzol Reagent (Invitrogen, Waltham, MA, USA) according to the manufacturer’s instructions. RNA concentration and purity were measured with NanoDrop.

### 4.4. Cloning of Human αB Crystallin

RNA isolated from skeletal muscle was treated with DNase I (Sigma D5307), and cDNA was retrotranscribed using SuperScript™ II Reverse Transcriptase (Invitrogen) using an Oligo(dT) primer according to the manufacturer’s indications. cDNA was used as a template to amplify αB crystallin CDS via PCR using PfuTurbo DNA Polymerase (Agilent) and the primers 5′-ATGGACATCGCCATCCACC-3′ and 5′-CGCAGCCCCCAAGAAATAG-3′. The resulting PCR product was then cloned into Zero Blunt TOPO PCR Cloning Kit (Invitrogen) following the manufacturer’s instructions to obtain the pTOPO-*CRYAB* vector.

### 4.5. Zebrafish Maintenance

Zebrafish AB wildtype strain and the previously generated *Tg(BMPRE:EGFP)ia18* transgenic line [33] have been maintained in the Facility of the University of Brescia at 28.5 °C in aerated saline water, under a 14 h light–10 h dark cycle, according to standard protocols [32]. For mating, males and females have been separated in the late afternoon and the next morning freed to start courtship, which ends with egg deposition and fertilization. Eggs have been collected and maintained at 28.5 °C in fish water (0.5 mM NaH2PO4, 0.5 mM NaHPO4, 0.2 mg/L methylene blue, 3 mg/L instant ocean) in a Petri dish. All manipulations and experiments have been performed on zebrafish embryos and larvae until 120 h post-fertilization (hpf) and do not require any formal authorization according to both the Standard Operating Procedures directives of the Animal Care and Use Committee of the University of Brescia and the directives of Italian Ministry of Health. We have also followed ARRIVE 2.0 guidance [58].

### 4.6. Plasmid Preparation

pcDNA3-wt*MYOT* vector containing wildtype human myotilin was a kind gift from Prof. V. Nigro. Tol2-EF1a promoter-EGFP and pBluscript-αactpromoter EGFP were kind gifts from Dr. Shaojun Du.

For this study, the plasmids Tol2-αactpromoter-EGFP, Tol2-αactpromoter-3XFLAG, Tol2-MYOT-αactpromoter-3XFLAG, Tol2-*CRYAB*-αactpromoter-3XFLAG, Tol2-*MYOT*S95I-αactpromoter-3XFLAG, and Tol2-*CRYAB*G154S-αactpromoter-3XFLAG were generated as described in the following.

The Tol2-EF1a promoter-EGFP vector was modified to replace the EF1a promoter with the skeletal-muscle-specific αactin promoter. The αactin promoter was amplified using PCR reaction from pBluscript-αactpromoter EGFP using PfuTurbo DNA Polymerase (Agilent) and the primers 5′-TAATACGACTCACTATAGGG-3′ and 5′-CCTTGGTCTGTGCAGGACA-3′. The resulting PCR product was subcloned into pTOPO plasmid using Zero Blunt TOPO PCR Cloning Kit (Invitrogen) following the manufacturer’s instructions to obtain the pTOPO- αactpromoter vector.

Then, the EF1A promoter on the Tol2- EF1A promoter-EGFP vector was replaced with the αactin promoter to generate the Tol2-αactpromoter-EGFP vector, as briefly described hereafter. The αactin promoter was amplified from the pTOPO-αactpromoter vector via PCR using PfuTaq polymerase (Agilent Technologies). The forward primer was 5′-TTGGGCCCGGCTCGAG TAATACGACTCACTATAGGG-3′, which includes an overhang complementary to the destination plasmid and recreates a XhoI site upstream of the promoter. The reverse primer was 5′-GACCTGCAGGAAGCTT CCT TGG TCT GTG CAG GACA-3′ that has overhangs complementary to the destination plasmid and recreates a HindIII site downstream of the promoter. The resulting PCR product was cloned into the Tol2- EF1A promoter-EGFP vector, previously digested with XhoI and HindIII (Thermofisher Scientific) to eliminate EF1A promoter, using In-Fusion HD Cloning Kit (Clontech) following the manufacturer’s instructions to obtain Tol2- αactpromoter-EGFP vector.

EGFP in Tol2- αactpromoter-EGFP vector was replaced with a smaller tag consisting of three repeats of the FLAG tag to generate the Tol2-αactpromoter-3XFLAG vector, as described below. Two oligonucleotides with a complementary sequence encoding three repeats of the FLAG tag were purchased from Primm (San Raffaele Biomedical Science Park, Italy). The oligonucleotide sequences were 5′-ACCGCGGTGGCGGCCGCTTACTTGTCATCGTCATCCTTGTAGTCGATGTCATGATCTTTATAATCACCGTCATGGTCTTTGTAGTCAGAATTCTTTGCCAAAA-3′. Besides the three repeats of the FLAG tag, the oligonucleotides include overhangs at both 5′ and 3′ ends that are complementary to the linearized destination plasmid and contain sequences to recreate EcoRI and NotI sites upstream and downstream of FLAG, respectively. The oligonucleotides include a stop codon after the three FLAG sequences. The oligonucleotides were mixed in a ratio 1:1 and incubates at RT for 30 min to allow hybridization. The resulting product was cloned into the Tol2-αactpromoter-EGFP vector, previously digested with EcoRI and NotI (Thermofisher Scientific) to eliminate EGFP, using the In-Fusion HD Cloning Kit (Clontech) following the manufacturer’s instructions to obtain Tol2- αactpromoter-3XFLAG.

The CDS for *MYOT* and *CRYAB* were subcloned into the Tol2-αactpromoter-3XFLAG vector. Briefly, CDS for *MYOT* and *CRYAB* were amplified using PCR with PfuTaq polymerase (Agilent Technologies) from pcDNA3-*MYOT* and pTOPO-*CRYAB,* respectively.

The forward primers used were 5′-TTTTGGCAAAGAATTCATGTTTAACTACGAACGTCCAA-3′ and 5′-TTTTGGCAAAGAATTCATGGACATCGCCATCCACC-3′ for myotilin and αBcrystallin, respectively. The forward primers include overhangs complementary to the linearized destination plasmid and recreate an EcoRI site upstream of CDS. The reverse primers were 5′-TTTGTAGTCAGAATTAAGTTCTTCACTTTCATAGAGTC-3′ and 5′-TTTGTAGTCAGAATTTTTCTTGGGGGCTGCGGT-3′ for myotilin and αBcrystallin, respectively. Reverse primers were designed to exclude the stop codon on the CDSs, have overhangs complementary to the linearized destination plasmid, and introduce an EcoRI site downstream of CDS. The resulting PCR products were cloned upstream of the 3XFLAG into the Tol2-αactpromoter-3XFLAG vector previously linearized with EcoRI (Thermofisher Scientific) using the In-Fusion HD Cloning Kit (Clontech) following the manufacturer’s instructions to obtain vectors Tol2-*MYOT*-αactpromoter-3X FLAG and Tol2-*CRYAB*-αactpromoter-3X FLAG.

The mutation n.564G>T in *MYOT* CDS (p.S95I, from now on referred to as *MYOT* S95I) and the mutation n.460G>A in *CRYAB* CDS (p.G154S, from now on referred to as *CRYAB* G154S) were introduced using QuikChange Site-Directed Mutagenesis Kit (Agilent) according to manufacturer’s instructions. To generate n.564G>T mutation in *MYOT* CDS the following primers were used: 5′-CAGTCCCCAGCCATCTTCCTCAGCTCC-3′ and 5′-GGAGCTGAGGAAGATGGCTGGGGACTG-3′.

To generate n.460G>A mutation in *CRYAB* CDS, the following primers were used: 5′-ACCAAGGAAACAGGTCTCTAGCCCTGAGCG-3′ and 5′-CGCTCAGGGCTAGAGACCTGTTTCCTTGGT-3′.

After every cloning and site-directed mutagenesis step, the sequence of the inserts/mutations was verified using Sanger sequencing.

The vectors Tol2-*MYOT*-αactpromoter-3XFLAG, Tol2-*CRYAB*-αactpromoter-3XFLAG, Tol2-*MYOT*S95I-αactpromoter-3XFLAG, Tol2-*CRYAB*G154S-αactpromoter-3XFLAG were injected in zebrafish embryos.

### 4.7. Plasmid Injection and Dose–Response Analysis

To choose the appropriate dose of the previously mentioned plasmids, we performed a dose–response analysis by injecting wildtype fertilized eggs at 1-cell stage with 4 nl of sterile water solutions containing Tol2 expression plasmids at concentrations of 20, 25, 40, 50, 75, and 100 ng/μL and phenol red (Merck KGaA, Darmstadt, Germany) to trace injection. Phenol red positive fertilized eggs have been selected 5 h post-injection by eliminating both unfertilized and phenol red negative eggs. By checking both embryo morphology and survival at 24 hpf, the final concentration of 25 ng/μL for each plasmid has been selected to perform all experiments of this study.

### 4.8. Survival Analysis

Kaplan–Meier analysis has been performed to check the effect of both wildtype and mutated *hMYOT* and *hCRYAB* isoforms on embryo survival at the early stage of development. We have recorded the number of dead embryos every 24 h until 7 dpf for each condition.

### 4.9. Tail Flip Analysis

Spontaneous coiling events, also called tail flips, are the first motor activity observable in Zebrafish embryo development. It is mediated by muscle innervation and originates in primary motor neurons of the parasympathetic system [59]. We have recorded the number of tail flips performed in 30 s by zebrafish embryos at 24 hpf for all the conditions.

### 4.10. Touch-Evoked Escape Response

The touch-evoked escape response test has been performed at 48 hpf to evaluate the response of embryos injected with 25 ng/μL of each plasmid to mechanical stimuli, which are linked to fast muscle fiber contraction [60]. After an external mechanical stimulus, embryo movements have been observed, and a score of 0–3 has been assigned. In total, 0 was assigned to completely paralyzed embryos; 1 was assigned to embryos performing only spontaneous coiling events; 2 was assigned to embryos moving short distances; and 3 was assigned to normally swimming embryos [17,61].

### 4.11. Birefringence

To check the integrity of muscle fiber structure with a non-invasive technique, we performed birefringence analysis on zebrafish embryos at 48 hpf anesthetized with tricaine methanesulfonate. This assay is focused on muscle fiber anisotropy which is the ability of organized muscle fibers to refract polarized light [30,62]. Briefly, a first filter polarized the microscope light. Polarized light ray passes through the anisotropic muscle fibers and is refracted. A second polarizer filter analyses the refraction angle. Thus, the birefringent light of normal muscle fiber appears as a brilliant signal. while deranged muscle fiber appears dark [63].

### 4.12. Transmission Electron Microscopy on Zebrafish

Zebrafish embryos at 48hpf from each condition have been fixed in 2.5% glutaraldehyde in 0.1 M phosphate buffer, pH 7.4 overnight at 4 °C. The embryos have been post-fixed with 1% osmium tetroxide and 1.5% potassium ferrocyanide at RT for 2 h, dehydrated with acetone and embedded in Epon 812 resin. Semithin sections (2 μm thick) were cut and stained with a 20% aqueous solution of toluidine blue. Skeletal muscle areas were trimmed, and ultrathin sections (70 nm thick) were cut and stained 1% Reynolds’ lead citrate.

### 4.13. Whole-Mount Immunofluorescence

Zebrafish embryos at 48 hpf from each condition have been fixed for 3 h in 4% PFA (Merck KGaA, Darmastadt, Germany) in PBS and then rehydrated by sequential 1 h washings with decreasing concentrations of methanol (from 100% to 25%) in PBS supplemented with 0.1% Tween 20 (PBT). Successively, embryos were permeabilized for 40 min with 10 μg/mL proteinase K in PBT and briefly re-fixed in 4% PFA in PBS for 15 min. After 3 PBT washes, embryos were incubated in 1% DMSO in PBT for 20 min at RT followed by two PBT washes of 10 min each. Embryos were then incubated in blocking solution (2% BSA, 2% heat-inactivated FBS, 1% DMSO in PBT) O.N. at +4 °C. After blocking, embryos were incubated for 48 h at +4 °C with the primary antibody diluted in blocking solution. To detect myotilin was used the NCL-Myotilin Ab clone RSO34 (Leica Biosystems Nussloch GmbH, Nußloch, Germany) at a final concentration of 1:150; to detect αB-crystallin was used the ABN185 Ab (Merck KGaA, Darmastadt, Germany) at a final concentration of 1:250. After Ab incubation and 3 washes of 10 min with PBST +1%DMSO, embryos have been finally incubated with secondary fluorescent antibodies (Anti-rabbit 488 produced in goat, Thermo Fisher Scientific Inc., Waltham, MA, USA; Anti-mouse 488 produced in donkey, Thermo Fisher Scientific Inc., Waltham, MA, USA) diluted 1:500 in blocking solution O.N. at 4 °C. After several washes in PBT, samples have been analyzed using both an Axiozoom.V16 fluorescence stereomicroscope equipped with Axiocamera 506 (Zeiss International, Oberkochen, Germany) and a Lightsheet Z1 microscope (Zeiss International, Oberkochen, Germany). 

### 4.14. Statistics

The statistical analyses were performed with GraphPad Prism 8 software, comparing non-injected animals to their injected siblings. Concerning the survival rate, the Kaplan–Meier test was carried out to verify whether the injection of all the constructs was able to alter the mortality rate. Comparison between analyzed samples has been performed by performing a non-parametric Wilcoxon-Mann–Whitney test. For the statistical evaluation of the tail flip, touch-evoked escape response was performed with one-way ANOVA with the Bonferroni correction. Data are reported as mean ± standard error of the mean. In all the analyses, the statistical significance was fixed at *p* ≤ 0.05.

## 5. Conclusions

In this manuscript, we demonstrated that transgenic zebrafish, transiently overexpressing both wildtype and mutant *hMYOT* and *hCRYAB* gene isoforms, is a powerful tool to study both gene function and their role in the pathogenesis of MFM. Further, our results suggest the importance of planning the future generation of stable zebrafish models harboring human mutations by transgenesis or CRISPR/Cas9 in the perspective of going towards Precision Medicine to obtain specific drugs for specific mutations and, thus, specific patients.

## Figures and Tables

**Figure 1 ijms-24-11483-f001:**
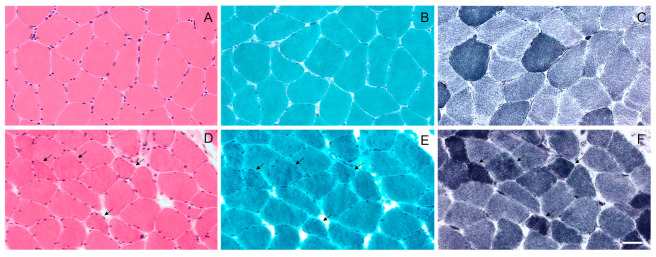
Light microscopy of the muscle biopsy from control subject (**A**–**C**) and patient with p.Gly154Ser *CRYAB* mutation (**D**–**F**). No morphological abnormalities with haematoxylin and eosin (**A**), modified Gomori trichrome (**B**) and NADH (**C**) staining in control subject. Haematoxylin and eosin (**D**) staining shows fiber size variation in MFM patient. Muscle fibers with amorphous material that stains eosinophilic on haematoxylin and eosin (**D**) and dark blue on modified Gomori trichrome (**E**) are also observed (arrows); these fibers have irregular focal areas of decreased or increased NADH staining (**F**) (arrows). Images were obtained with obj ×20. Scale bar: 50 μm.

**Figure 2 ijms-24-11483-f002:**
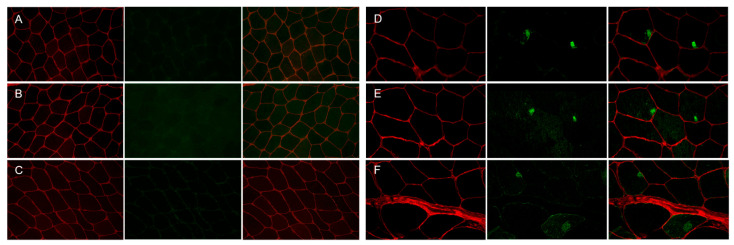
Immunofluorescence staining of the muscle biopsy from control subject and patient with p.Gly154Ser *CRYAB* mutation. Panels (**A**–**C**) (control), Panels (**D**–**F**) (patient). No focal areas of increased reactivity for αB-crystallin (Panel (**A**), green), desmin (Panel (**B**), green) and myotilin (Panel (**C**), green) in muscle fibers of control muscle. Focal accumulation of αB-crystallin (Panel (**D**), green), desmin (Panel (**E**), green) and myotilin (Panel (**F**), green) in two fibers of patient’s muscle biopsy. The percentage of fibers with aggregates is less than 15% of the total muscle fibers, and the percentage of the fiber area occupied by aggregates in the sections analyzed is 10.03 ± 2.15 for αB-crystallin, 5.78 ± 1.30 for desmin and 10.25 ± 0.81 for myotilin of the total muscle fiber area in this patient. Maackia amurensis lectin-II (MAL-II) staining (Panels (**A**–**F**), red) depicts the sarcolemma of muscle fibers. Images were obtained with obj ×20.

**Figure 3 ijms-24-11483-f003:**
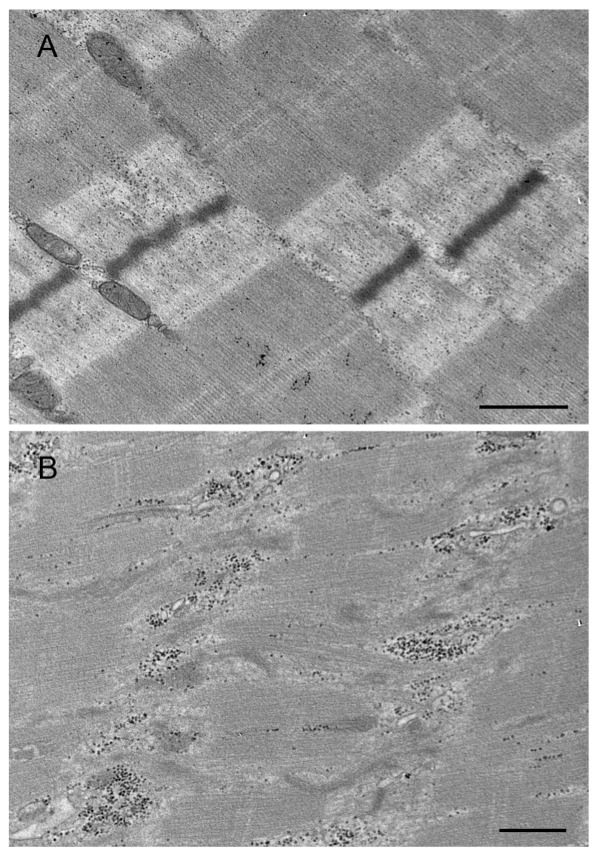
Transmission electron micrographs of the muscle biopsy from control subject (**A**) and patient with p. Arg259Cys *MYOT* mutation (**B**). Normal muscle sarcomeres in muscle of control subject (**A**). An abnormal fiber region showing myofibrillar disruption and streaming of Z-disk in patient (**B**). Scale bar: 500 nm.

**Figure 4 ijms-24-11483-f004:**
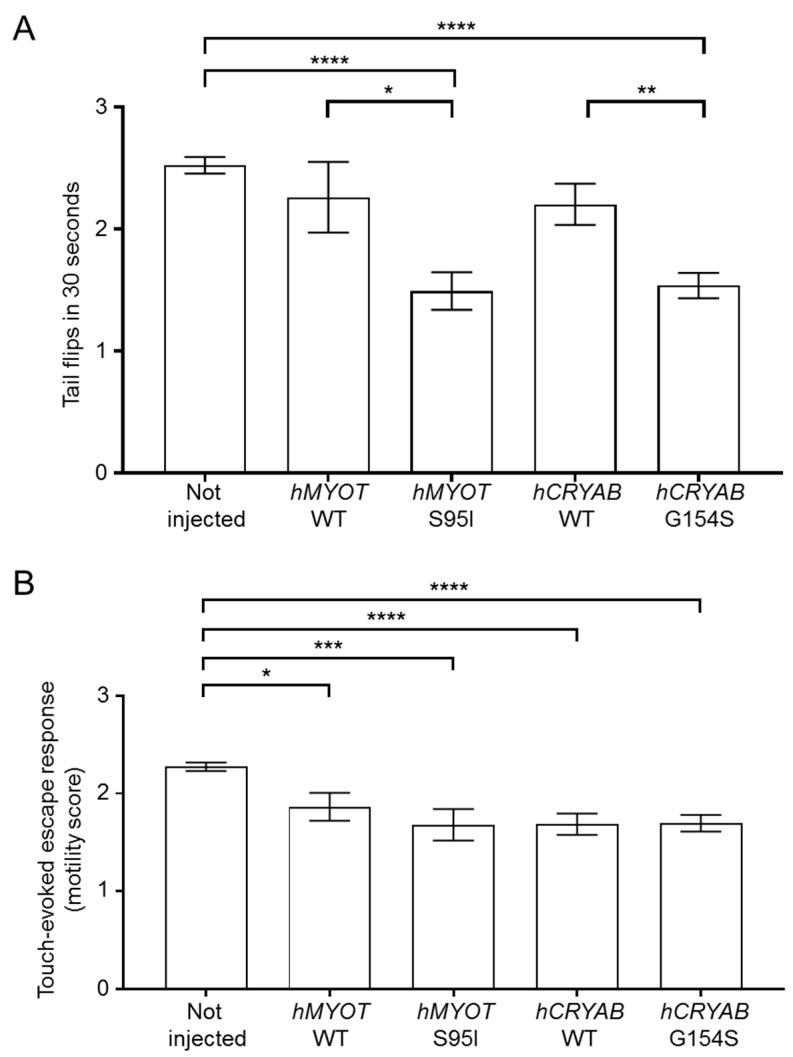
Effects on zebrafish motor behavior of both wildtype and mutant *hMYOT* and *hCRYAB* genes. (**A**). Tail flips. Spontaneous coiling events (tail flips) performed by wildtype zebrafish (not injected or injected with 25 µM of tol2 expression plasmid harboring *hMYOT* WT, *hMYOT* S95I, *hCRYAB* WT, *hCRYAB* G154S cDNA sequences) were recorded at 24 hpf. Graph bars represent the media of tail flips performed by each embryo in 30 s ± s.e.m. (**B**). Touch-evoked escape response was measured at 48 hpf on the same embryos reported in (**A**). A value of 0 was attributed to completely paralyzed embryos, 1 to embryos performing only spontaneous coiling events, 2 to embryos moving short distances, and 3 to embryos swimming normally. Graph bars represent the media of motor value for each condition ± s.e.m. Values in (**A**,**B**) represent the media from 4 independent experiments. Number of embryos analyzed for (**A**): not injected (n = 330), *hMYOT* WT (n = 50), *hMYOT* S95I (n = 59), *hCRYAB* WT (n = 109), *hCRYAB* G154S (n = 168). Number of embryos analyzed in (**B**): not injected (n = 342), *hMYOT* WT (n = 66), *hMYOT* S95I (n = 50), *hCRYAB* WT (n = 108), *hCRYAB* G154S (n = 148). *p* values were calculated by using One-way ANOVA with Bonferroni correction. * *p* < 0.05; ** *p* < 0.01; *** *p* < 0.001; **** *p* < 0.0001.

**Figure 5 ijms-24-11483-f005:**
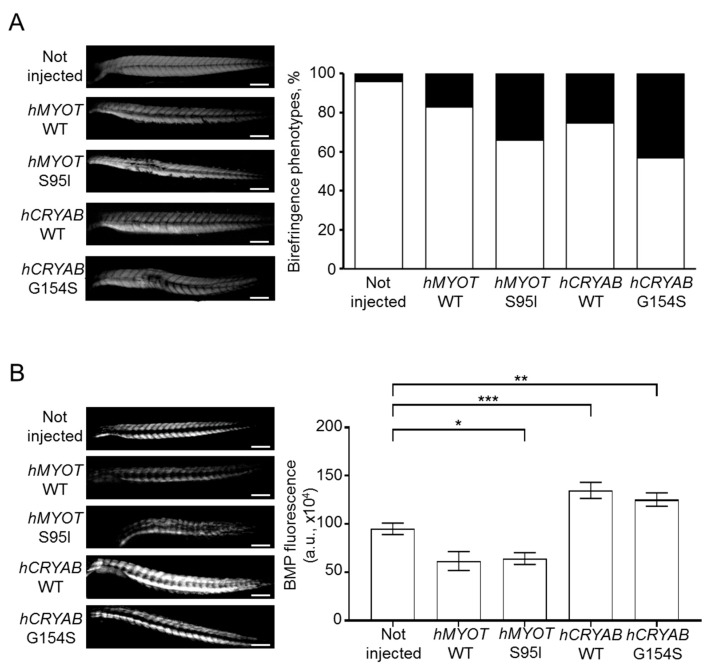
Effects on muscle fiber structure and development of both wildtype and mutant *hMYOT* and *hCRYAB* genes. (**A**). Birefringence analysis of zebrafish embryos at 48 hpf. Left panels in (**A**) show representative birefringence images of embryos for each analyzed condition. Scale bar is 200 μm. Total number of embryos analyzed for each condition: not injected wildtype embryos (n = 199), injected embryos with *hMYOT* WT (n = 35), *hMYOT* S95I (n = 41), *hCRYAB* WT (n = 87), *hCRYAB* G154S (n = 92). Right panel in (**A**) shows the bar graph reporting the percentage of birefringence phenotypes for each analyzed condition. By considering 100% as the total number of embryos analyzed for each condition, the white open bars represent the percentage of embryos with normal birefringence phenotype, while the closed black bars represent the percentage of severe birefringence phenotype. Three independent experiments have been performed. (**B**). The involvement of BMP signaling pathway in muscle structure development was evaluated by checking GFP fluorescence of Tg(BMPRE:EGFP)ia18 zebrafish embryos at 48hpf not injected (n = 127) or injected with *hMYOT* WT (n = 32), *hMYOT* S95I (n = 57), *hCRYAB* WT (n = 68), *hCRYAB* G154S (n = 98). Left panels in (**B**) show representative images of fluorescent embryos for each analyzed condition. Scale bar is 200 μM. Right panel in (**B**) shows the bar graph reporting the integrated density of fluorescent pixel area in the trunk region ± s.e.m.. Four independent experiments were performed. * *p* < 0.05, ** *p* < 0.01, *** *p* < 0.001 according to One-way ANOVA test with Bonferroni correction.

**Figure 6 ijms-24-11483-f006:**
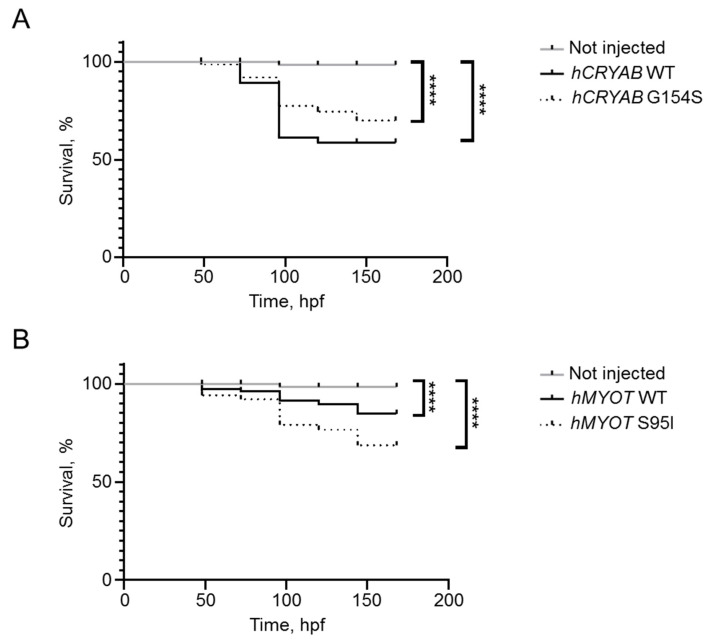
Effects on fish survival of both wildtype and mutant *hCRYAB* (**A**) and *hMYOT* (**B**) genes. Kaplan–Meier analysis was performed to observe the number of survived fish up to 7 dpf. Graphs in (**A**,**B**) reported the percentage of survived embryos after 4 experiments. Number of fish in (**A**,**B**): not injected (n = 112); *hCRYAB* WT (n = 63); *hCRYAB* G154S (n = 111); *hMYOT* WT (n = 73); *hMYOT* S95I (n = 79) of injected fish. **** *p* < 0.0001 according to non-parametric Wilcoxon-Mann–Whitney test.

**Figure 7 ijms-24-11483-f007:**
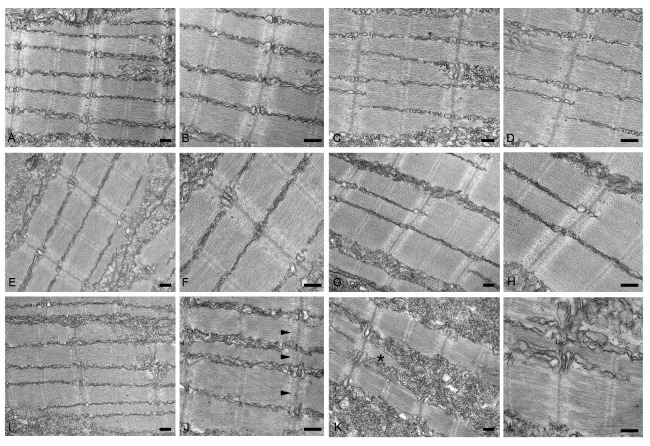
Transmission electron micrographs of no-injected (**A**,**B**) or injected with vehicle (**C**,**D**), *hMYOT* WT (**E**,**F***), hCRYAB* WT (**G**,**H**), *hMYOT* S95I (**I**,**J**), *hCRYAB* G154S (**K**,**L**). In non-injected wildtype embryos or injected embryos with vehicle or wildtype genes muscle fibers show normal sarcomere structure with highly organized and well-aligned thick and thin myofilaments flanked by Z-disks. By contrast, embryos injected with *hMYOT* S95I sarcomeres showed abnormalities of Z-disk (black arrowheads in panel (**J**)), which appears irregular with its extension into I band. Large areas depleted of myofilaments, stressed fiber-like structures (black asterisk in panel (**K**)) and reduced density of the sarcomeric myofilaments were observed in myofibers of *hCRYAB* G154S injected embryos. Scale bar, 200 nm.

**Figure 8 ijms-24-11483-f008:**
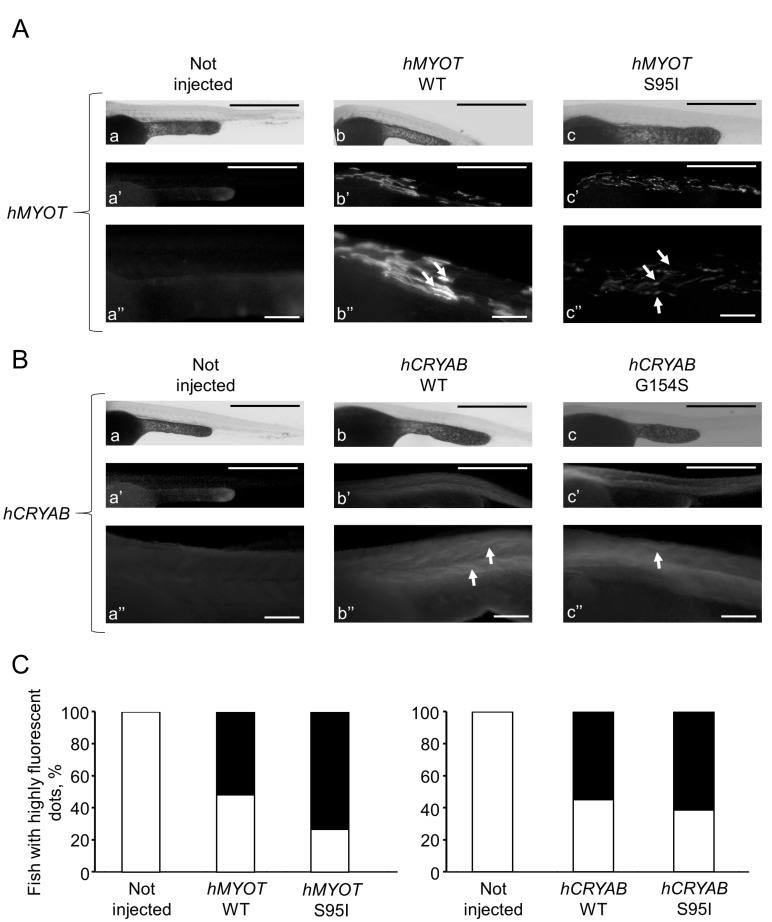
Whole-mount immunofluorescence shows highly fluorescent dots in zebrafish embryos injected with both wt and mutated *hCRYAB* and *hMYOT.* Whole-mount immunofluorescence analysis with specific myotilin and αb-crystallin antibodies was performed at 48 hpf, and images of muscle fibers in the trunk region were acquired with Axiozoom.V16 fluorescent stereomicroscope equipped with Axiocamera 506 (Zeiss International, Oberkochen, Germany). In (**A**,**B**), representative full Z-stack brightfield and fluorescent images are reported for zebrafish wildtyp*e* embryos not injected (**a**–**a″** in **A**,**B**) or injected with *hMYOT* WT (**b**–**b″** in **A**), *hMYOT* S95I (**c**–**c″** in **A**), *hCRYAB* WT (**b**–**b″** in **B**), *hCRYAB* G154S (**c**–**c″** in **B**). Panels (**a″**, **b″** and **c″**) both in (**A**,**B**) represent zoomed images from panels (**a**, **b** and **c**), respectively. Scale bar is 500 μm for (**a**,**a′**,**b**,**b′**,**c**,**c′**) panels and 100 μm for (**a″**,**b″**,**c″**) panels. Highly fluorescent dots of different shapes and sizes (white arrows) are indicated. (**C**). Bar graphs report the percentage of fish showing highly fluorescent dots (black closed bars). White open bars represent fish showing no fluorescent dots. Three independent experimental analyses have been performed. Total number of embryos analyzed for each experimental condition: wildtype not injected (n = 30 in (**A**) and left panel in (**C**); n = 35 in (**B**) and right panel in (**C**)), *hMYOT* WT (n = 30), *hMYOT* S95I (n = 56), *hCRYAB* WT (n = 20), *hCRYAB* G154S (n = 26).

**Figure 9 ijms-24-11483-f009:**
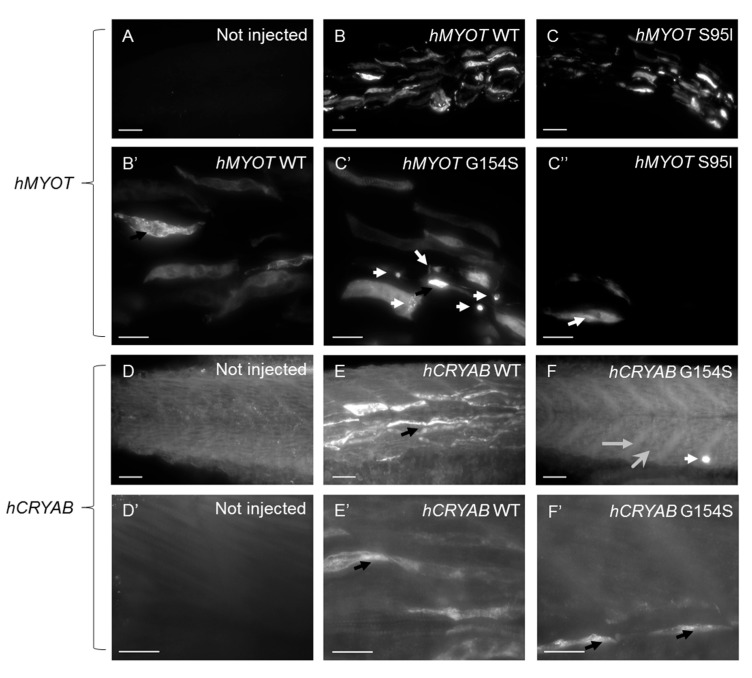
Lightsheet microscopy shows highly fluorescent dots in zebrafish embryos injected with both wt and mutated *hCRYAB* and *hMYOT*. Whole-mount immunofluorescence analysis with specific myotilin and αb-crystallin antibodies was performed at 48 hpf, and images of muscle fibers in the trunk region were acquired with a Lightsheet Z1 microscope (Zeiss International, Oberkochen, Germany). Images are Maximum Intensity Projections of selected planes for zebrafish wildtype embryos not injected (**A**,**D**,**D′**) or injected with *hMYOT* WT (**B**,**B′**), *hMYOT* S95I (**C**,**C′**,**C″**), *hCRYAB* WT (**E**,**E′**), *hCRYAB* G154S (**F**,**F′**). Total number of embryos analyzed for each experimental condition n = 3. Three independent experimental analyses have been performed. Panels (**B′**,**C′**,**C″**,**D′**,**E′**,**F′**) represent zoomed images from panels (**B**,**C**,**D**,**E**,**F**), respectively. Scale bar is 50 μm for (**A**–**F**) panels and 20 μm for (**B′**,**C′**,**C″**,**D′**,**E′**,**F′**) panels. Highly fluorescent dots variable in shape and size (black and white arrows), extended myotendinous junction (grey arrow in (**F**)) and deranged muscle fibers (large gray head arrow in (**F**)) are indicated.

## Data Availability

The data supporting the findings of this study are contained within the contents of this article. The datasets generated during this study will be freely provided by the corresponding author upon request.

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
