# Peer review of "Human Mutated MYOT and CRYAB Genes Cause a Myopathic Phenotype in Zebrafish"

_ijms, 2023, doi:10.3390/ijms241411483_

Round 1

Reviewer 1 Report (Previous Reviewer 2)

The study described in this manuscript focuses on investigating the role of specific gene mutations in myofibrillar myopathies (MFM) using zebrafish as a model organism. MFM is a group of neuromuscular disorders characterized by protein misfolding, aggregation, and muscle weakness.

The researchers aimed to understand the pathogenesis of MFM by studying the effects of pathogenic mutations in the myotilin (MYOT) and αB-crystallin (CRYAB) genes, which are known to be associated with MFM, in zebrafish muscle tissue.

The study involved the overexpression of wild-type and mutant forms of the MYOT and CRYAB genes in zebrafish embryos. The researchers then examined the structural and functional changes in the zebrafish skeletal muscle caused by these gene mutations.

Histological analysis revealed significant reductions in myofiber density, disorganization of sarcomeres, and the formation of granular protein aggregates in the zebrafish muscle tissue, resembling the histopathological features observed in human MFM.

Additionally, the study investigated the effects of gene overexpression on fish survival and found a progressive decrease in survival, particularly with the overexpression of the CRYAB gene.

The researchers concluded that the zebrafish models overexpressing the mutant MYOT and CRYAB genes provide a valuable tool for studying the function of these genes and their role in the pathogenesis of MFM. They also emphasized the importance of developing stable zebrafish models harboring human mutations for future research and the potential for precision medicine approaches in treating MFM.

Overall, the study contributes to the understanding of the molecular mechanisms underlying MFM and highlights the potential of zebrafish models for studying neuromuscular disorders and identifying potential therapeutic targets.

-       How did the authors make sure that the insertion of their plasmid was successful? Did the authors inject a fluorescent reporter gene, or conducted a PCR or qPCR or an in situ hybridization? If they did so, the authors should mention it in their manuscript?

-       Did the insertion of MYOT wild type gene and the CRYAB wild type gene induce any toxicity do the zebrafish? The authors mentioned that they assessed the toxicity, but I am interested to see of the insertion of WT human genes have an effect on the zebrafish. I am asking this question because if we compare the zebrafish with and without the injection of human genes, there is a difference (even if it is not significant) in all the experiments that the authors presented. Thus, can the authors please elaborate on this matter?

-       There is a difference in the BMP expression between the non-injected zebrafish and the zebrafish injected with the MYOT WT gene. Yet, the authors shows that this difference is not significant. It is weird because by observing the error bars (SEM) there is an important difference between these two groups. Can the author elaborate?

-       Why did the CYRAB WT gene induced more mortality in the zebrafish compared to the CRYAB mutated gene? And why both CRYAB WT and MYOT genes induced mortality to zebrafish?

-       It would have been interesting if the authors have analyzed the level of desmin, myotilin and αβ-crystallin in the zebrafish that were injected with MYOT and CRYAB WT genes and the mutated genes as well to assess if they found the same results as the ones obtained with the patients from their cohort study.

-       As much as I am interested by the results presented by the authors regarding the implication of the MYOT and CRYAB genes in the MFM, I think that the insertion of the WT gene to the zebrafish is inducing a pseudo MFM profile and thus I don’t know if the zebrafish is the most convenient model for this kind of study. Can the author elaborate on this matter?

-       What implications do your findings have for the development of precision medicine approaches for myofibrillar myopathies? 

-       Remove all the yellow highlight from the text.

-       The phrase from line 44 to line 47 should be revised. If the authors are stating that that the abnormal fibers are the main morphological hallmark and that they are positive for several proteins, including αB-crystallin, desmin, and myotilin, then I suggest that the phrase should be: “The diagnosis is established by muscle biopsy, which shows abnormal fibers containing amorphous material of irregular shape and size that are positive for several proteins, including αB-crystallin, desmin, and myotilin, as the main morphological hallmark.”

-       Please revise the phrase from line 79 to 82. It is long and quite confusing.

Author Response

Reviewer 2 Report (New Reviewer)

This paper, appropriately called a pilot study, is a good first step towards the author's goal of identifying useful treatments for a rare human muscular disorder. The research group is expert in the use of zebrafish as a research model. The design is sound and the methods appear to be carefully conducted. The conclusions are supported by the data and are limited by the bounds of a pilot study. A few minor English corrections are needed. Two words in the Abstract have hyphens and the word also in line 156 might be omitted.

The English quality is good. There are minor corrections needed, listed in the comments for Authors above.

Author Response

This manuscript is a resubmission of an earlier submission. The following is a list of the peer review reports and author responses from that submission.

Round 1

Reviewer 1 Report

Cannone et al. utilize MYOT S95I and CRYAB G154S-expressing zebrafish to study myofibrillar myopathies. While the intention of this study was interesting, the experimental design lacks rigor; necessary controls are lacking, many results are not quantified, and many conclusions are not convincing. Furthermore, the results offer no real conclusion that furthers knowledge of the pathogenesis or treatment of myopathies.

1.     The percentages in lines 75-78 are misleading. They represent the numbers of each mutation out of the 10 MFM cases. Which means that the total number of MFM cases with those mutations are significantly lower.

2.     It is unclear why figures 1-3 are not described in the text.

3.     The muscle sections in Figure 1 look normal, there’s no obvious myopathy. The authors should include a healthy patient muscle as a control for these histological stains.

4.     A healthy patient muscle should be used a control in Figures 2 and 3. Moreover, in Figure 2, outline of the muscle fiber should be demarcated by staining for proteins like WGA. Also, quantifications of these focal accumulations per fiber area should be shown.

5.     Instead of knocking in the mutations the authors overexpress the human MYOT or CRYAB gene that harbors the mutation. There is no explanation provided that justifies this less robust technique.

6.     The authors should explain the experimental design in Figure 5A more carefully. It is unclear whether the phenotype presented is significant. It is also unclear what the open and closed boxes as well as the y-axis represent.

7.     The rationale to investigate BMP signaling is poor. The authors make an unjustified conclusion in lines 141-142 because BMP fluorescence contrastingly decreases in hMYOT WT and mutant zebrafish. The authors fail to comment on this observation. Images of immunostaining for GFP should be included as part of this figure.

8.     The impaired survival for hCRYAB G154S described in Figure 6 could merely be by chance. It is unclear why zebrafish expressing the WT constructs also have impaired survival. This result provides further evidence that a knock-in strategy should have been used to study MFM in zebrafish.

9.     Immunostaining in Figure 7 should be quantified. Furthermore, brightfield images of the zebrafish are necessary.

Reviewer 3 Report

The authors injected plasmids expressing the wild type and mutated proteins hMYOT and hCRYAB into one-cell stage zebrafish embryos to investigate the role of myotilin and αB-crystallin in the pathogenesis of myofibrillar myopathies. In my opinion, exploring the functions of target proteins through injecting plasmids into zebrafish embryos is unacceptable due to toxicity of exogenous DNA and mosaic expressing of the target genes. As stated by the authors the injected plasmids demonstrated high toxicity in zebrafish embryos. Why didn’t they use gene knockout or knock-in zebrafish lines? Furthermore, injecting plasmids expressing both wildtype and mutated hMYOT and hCRYAB gene isoforms resulted the same defects. I’m afraid the observed defects in the injected zebrafish embryos are caused by ectopic over expression of the human protein per se and have nothing to do with the mutation. As shown in Figure 2, the patient muscle fibers had focal accumulation of αB-crystallin, desmin, and myotilin. There is no convincing causal relationship between the mutation in the target genes and the observed defects of the injected zebrafish embryos.

Minor concerns:

Figure 1 is not well annotated. Is there any difference between A and D, B and E, C and F? Are there pictures for normal controls? The abnormal structures in the photographs should be indicated.

Figure 3 is not well annotated. The stated abnormal structure should be pointed using arrows.

Lines 75-80, association of the casual mutation in the listed genes with MFM is not clearly described. How were the mutations identified? Is there significant association between the mutations and MFM?

Round 2

Reviewer 3 Report

The rationale of using zebrafish to model human disease is based on the high genetic homology between these two species. The authors state that both the CRYAB and MYOT zebrafish genes share low degree of both nucleotide and aminoacidic sequence homology with the human counterparts (about 40 % in the functional domain of myotilin; about 60% in the functional domain of αB-crystallin). Furthermore, both the S95 (myotilin) and G154 (αB-crystallin) residues are not conserved in the zebrafish proteins. Therefore, the functions of S95 (myotilin) and G154 (αB-crystallin) are not conserved in zebrafish and both the wild type and mutated proteins are mutants for zebrafish. It can be reasoned the defects in the injected zebrafish larvae were caused by accumulation of the exogenous proteins in the muscle cells and had nothing to do with the mutation. This paper has no obvious theoretical significance in terms of unravelling the mechanisms underlying the pathology caused by the mentioned mutations.

Moreover, the authors call the plasmid-injected zebrafish embryos as “lines” and “mutants”. This is incorrect. Zebrafish lines or mutants indicate those carry stable and heritable genetic variations in the genome.